



# Miniaturised visible and near-infrared spectrometers for assessing soil health indicators in mine site rehabilitation

Zefang Shen[1], Raphael A. Viscarra Rossel[1], Haylee D'Agui[2], Lewis Walden[1], Mingxi Zhang[1], Tsoek Man Yiu[1], Kingsley Dixon[2], Paul Nevill[2,3], Adam Cross[2,4], Mohana Matangulu[1], and Yang Hu[1]

[1]Soil and Landscape Science, School of Molecular and Life Sciences, Faculty of Science and Engineering, Curtin University, G.P.O. Box U1987, Perth, WA 6845, Australia
[2]ARC Centre for Mine Site Restoration, School of Molecular and Life Sciences, Faculty of Science and Engineering, Curtin University, G.P.O. Box U1987, Perth, WA 6845, Australia
[3]Trace and Environmental DNA Laboratory, School of Molecular and Life Sciences, Faculty of Science and Engineering, Curtin University, G.P.O Box U1987, Perth, WA, 6845, Australia
[4]EcoHealth Network, 1330 Beacon St, Suite 355a, Brookline, MA 02446, United States. https://ecohealthglobal.org.

**Correspondence:** Raphael A. Viscarra Rossel (r.viscarra-rossel@curtin.edu.au)

**Abstract.** Mining can cause severe disturbances to the soil, which underpins the viability of terrestrial ecosystems. Post-mining rehabilitation relies on measuring soil properties that are critical soil health indicators. Soil visible–near-infrared (vis–NIR) spectroscopy is rapid, relatively accurate and cost-effective for estimating a range of soil properties. Recent advances in infrared detectors and microelectromechanical systems (MEMS) have produced miniaturised, relatively inexpensive spectrometers.

Here, we evaluate the spectra from four miniaturised visible and NIR spectrometers, some combinations and a full-range vis–NIR spectrometer to model 29 soil physical, chemical and biological properties used to assess soil health at mine sites. We collected soils from reference undisturbed native vegetation and topsoil stockpiles from seven mines in Western Australia. We evaluated the repeatability of the spectrometers and the accuracy of the spectroscopic models built with seven statistical and machine learning algorithms. The spectra from the visible spectrometer could estimate soil texture (sand, silt, and clay) more

accurately than the NIR spectrometers. However, the spectra from the NIR spectrometers produced better estimates of soil chemical and biological properties. By combining the miniaturised visible and NIR spectrometers, we improved the accuracy of their soil property estimates, which were similar to those from the full-range spectrometer. The miniaturised spectrometers and combinations predicted 24 of the 29 soil properties with moderate or greater accuracy (Lin's concordance correlation, $\rho_c \geq 0.65$). The repeatability of the NIR spectrometers was similar to that of the full-range, portable spectrometer. Our results

show that the miniaturised NIR spectrometers can produce accurate predictions of soil properties comparable to the (orders of magnitude) more expensive full-range portable system, particularly when combined with a visible range spectrometer. Thus, there is potential to develop rapid, accurate, cost-effective diagnostic capacity to support mine site rehabilitation based on miniaturised spectrometers and deliver significant positive economic and environmental outcomes.



## 1 Introduction

Soil health underpins the viability of terrestrial ecosystems, whether natural or human-altered (e.g., agricultural, post-mining) and provides a variety of functions essential for life on Earth (Jeffrey, 2017; Timmis and Ramos, 2021). Healthy soil supports both above- and below-ground biodiversity, as well as plant growth, agricultural productivity, and a suite of ecological functions and ecosystem services (Timmis and Ramos, 2021). Consequently, returning functional soils to a site following disturbance is critical to achieving sustainable and resilient rehabilitation (e.g., reinstating a level of ecosystem productivity or functioning)

or ecological restoration (e.g., assisting the recovery of an ecosystem that has been degraded, damaged, or destroyed; (Gann et al., 2019)). Some of the most severe disturbances to soil result from surface mining (Cooke and Johnson, 2002; Cross et al., 2017). Rehabilitation or ecological restoration is often a regulatory requirement for mining companies to undertake during mine closure (Manero et al., 2021). There is an increasing expectation that mining companies return functional, resilient and biodiverse native ecosystems to lands where mining has occurred. However, rehabilitation or ecological restoration can be

challenging on mined lands because the substrates generated by mining, such as tailings (fine particulate materials), waste rock, and salvaged topsoil (e.g. Stock et al. 2020), can be different from undisturbed homologues (Munoz-Rojas et al., 2016; Cross and Lambers, 2021; Cross et al., 2018). Additionally, different approaches to rehabilitation and ecological restoration practices such as landform design and contouring, topsoil return, seeding and planting, can significantly influence soil characteristics and other ecosystem attributes. Ensuring mined lands are placed on favourable ecological trajectories requires a fundamental

understanding of the edaphic conditions of the pre-disturbance landscape and mined materials and how these conditions might influence the properties of soil and its capacity to support ecological functioning (Cross et al., 2021).

Soil health, which we define as the capacity of soil to sustain biodiversity and biological productivity, and maintain ecological functioning and ecosystem services, represents an intricate series of interactions between important soil physical, chemical and biological properties (Lehmann et al., 2020). Soil physical properties are essential for the provisioning of air, water, gaseous

exchanges, and habitat; chemical properties for moderating soil reactions and nutrient transformations and availability; and biological properties for nutrient cycling (Lal, 2004). Soil health is fundamental to plant productivity and landscape stability (Turner et al., 2018). It is also essential to soil functioning (e.g., a medium for plant growth, habitat for soil organisms, carbon storage), which underpins most post-mining land uses, such as conservation and the reinstatement of native ecosystems to alternative uses such as agriculture. The assessment of soil health through examination of key physical, chemical, and biological

indicators can help to guide, monitor and evaluate ecological trajectory following restoration or rehabilitation (Cross et al., 2021; Rinot et al., 2019). Failure to understand and effectively manage soil health will likely result in undesirable trajectories with adverse and often cascading, long-term harmful impacts on biodiversity, ecosystem productivity and resilience.

Assessment of soil health in post-mining rehabilitation and ecological restoration, when undertaken, remains typically an analysis of key indicators (e.g., soil nutrient concentrations, pH, electrical conductivity, cation exchange capacity) on composite

samples collected from representative locations around a site. Assessment is generally undertaken periodically from the beginning of rehabilitation or ecological restoration activities to monitor changes in indicators over time, with collected soil samples analysed in a laboratory to provide average values of the soil properties. However, conventional assessment in this manner is



time-consuming and expensive because laboratory analyses require elaborate methods and specialised equipment and proce-
dures and can be prone to errors resulting from inappropriate sampling, transportation, preparation, or analysis (Viscarra Rossel
and Bouma, 2016). These constraints often result in the collection of only a few samples, limiting spatial and temporal repre-
sentation and the ability of sampling to characterise soil variability at a site adequately. Additionally, practitioners must also
wait for laboratory results to be returned following sample submission, limiting their ability to adapt management and interven-
tion activities rapidly. There is, therefore, a need for scientifically robust diagnostic capacity able to rapidly, cost-effectively,
and accurately quantify soil health via the assessment of key soil property indicators in the field.

The literature has shown that many key indicators of soil health, such as organic carbon, texture, water content, cation
exchange capacity, pH, microbial biomass and diversity, can be modelled with visible–near-infrared spectra (vis–NIR; 400–
2500 nm) (e.g. Viscarra Rossel et al., 2006; Stenberg et al., 2010; Guerrero et al., 2010; Soriano-Disla et al., 2014; Yang
et al., 2019). The modelling is possible because soil properties can be multivariately related to the wavelengths in the spectra,
which contain information on the inherent composition of the soil, which comprises minerals, organic compounds and water
(Viscarra Rossel and Behrens, 2010). Thus, these spectra can describe soil both qualitatively and quantitatively (Nocita et al.,
2015; Askari et al., 2015). Broad, weak absorptions at wavelengths smaller than 1000 nm can result from chromophores and
iron oxides; narrow, well-defined absorptions at wavelengths between 1400–1900 nm are due to hydroxyl bonds and water;
absorptions at wavelengths around 2200 nm occur from clay minerals; and organic matter absorbs in different regions through-
out the visible and NIR range. vis–NIR spectroscopy also provides information on soil particle size, and thus information on
the soil matrix (Stenberg et al., 2010). Hence, in addition to soil properties, soil type and soil horizons can also be determined
using vis–NIR spectra (Viscarra Rossel and Webster, 2011).

There are advantages to using the spectroscopic method. First, spectroscopic measurements are highly reproducible (Sten-
berg et al., 2010). Once spectroscopic models of soil properties are derived and validated, one can use them to estimate the
values of those properties where those measurements are lacking and would be too expensive to make using conventional
laboratory methods (Viscarra Rossel et al., 2006; Nocita et al., 2015). Spectroscopic models can be built with multivariate
regressions, such as partial least squares regression (PLSR) or machine learning methods such as support vector machines
(SVM), regression trees, neural networks (Viscarra Rossel and Behrens, 2010); or more recently, deep learning (e.g. Shen
and Viscarra Rossel, 2021). Second, large databases of soil spectra are being developed to help meet the growing demand for
soil information to evaluate and monitor soil at a range of scales (Viscarra Rossel and Webster, 2012; Orgiazzi et al., 2018;
Viscarra Rossel et al., 2016). Third, as technologies develop, spectrometers have become cheaper, smaller, portable, and more
accessible. Emerging infrared detector technologies are being used to produce miniaturised hand-held instruments that are
rugged and affordable, also using microelectromechanical structures (MEMSs) (Christian and Ford, 2021; Johnson, 2015),
thin-film filters, light-emitting diodes (LED), fibre optics, and high-performance detector arrays (Coates, 2014). The above
combined provide an opportunity to develop a needed quantitative soil health diagnostic capacity.

Here, we evaluate the spectroscopic method using one portable and four miniaturised spectrometers, estimating 29 soil
physical, chemical and biological properties considered to be indicators of soil health in the rehabilitation and ecological
restoration of mined land. We aimed to assess the spectrometers' repeatability and the accuracy of their estimates of the





soil properties using PLSR and several machine learning algorithms. We used undisturbed natural and stockpiled topsoils collected from seven mine sites across Western Australia, one of Australia's most significant mining regions (MCA, 2010). There are published studies that compared miniaturised spectrometers for calibrating soil properties (Tang et al., 2020; Ng et al., 2020; Soriano-Disla et al., 2017; Sharififar et al., 2019). However, our study is the first to evaluate portable, miniaturised and combinations of spectrometers with different algorithms for their capacity to estimate a diverse range of soil properties useful for mine site rehabilitation or ecological restoration.

## 2   Methods

Seven mine sites in Western Australia were selected as study sites, representing a range of climatic conditions, soil types, vegetation assemblages, and commodity types (Table 1, Figure 1).

**Table 1.** Resource type mined, predominant soil type, dominant vegetation type, and climate type for each of the mine sites included in the study. Note that 'Mine F' has requested non-disclosure of resource mined.

| Code | Resource Mined | Soil | Vegetation Type | Climate |
|------|----------------|------|-----------------|---------|
| A | Mineral Sands | Chromosols | Banksia woodland | Warm-summer mediterranean |
| B | Iron Ore | Tenosols | Savanna woodland | Hot desert |
| C | Iron Ore | Sodosols | Savanna woodland | Hot desert |
| D | Copper, Cobalt, Nickel | Calcarosols | Mallee and heath woodland | Warm semi-arid |
| E | Nickel | Tenosols | Open shrubland and grassland | Hot desert |
| F | (Undisclosed) | Podosols | Jarrah Forest | Warm-summer mediterranean |
| G | Bauxite | Kandosols | Jarrah Forest | Warm-summer mediterranean |

Sites included a mineral sands mine (Figure 1A); two iron ore mines (Figure 1B, C); a copper-cobalt-nickel mine (Figure 1D); a nickel mine (Figure 1E); a bauxite mine (Figure 1G), and a mine for which the commodity will remain undisclosed (anonymity requested; Figure 1F). The soil types are orders from the Australian Soil Classification (Isbell, 2002).





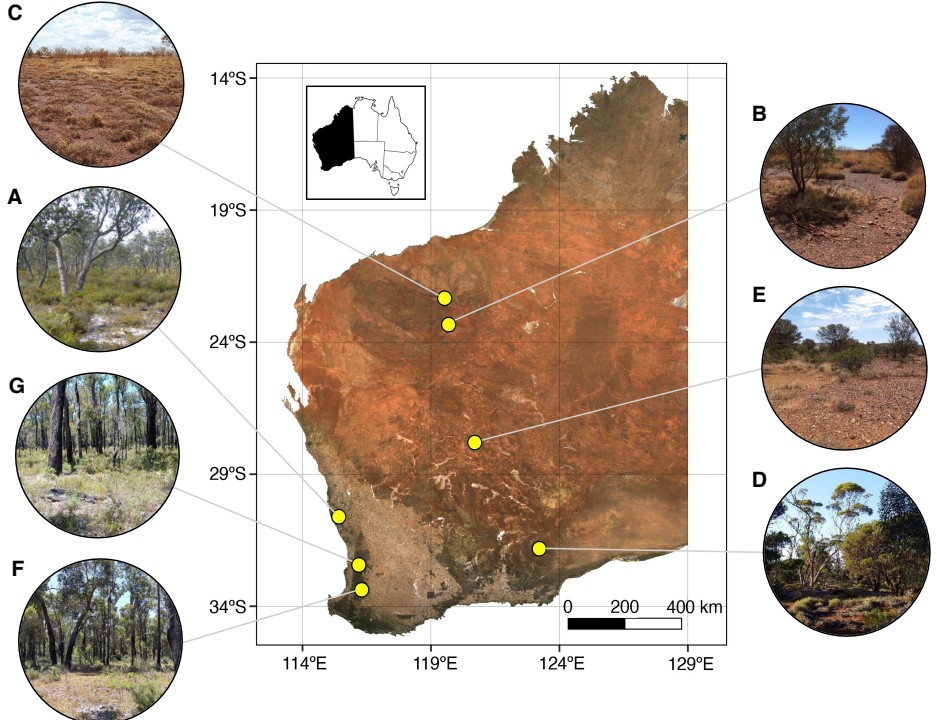

**Figure 1.** Location of mine sites within Western Australia from which soil samples were collected for spectroscopic analysis. Soils were collected from a variety of vegetation types in different climatic regions, including from (A) a mineral sands mine within banksia woodland in a Warm-summer mediterranean region; (B, C) two iron ore mines within savanna woodland in the Hot desert region; (D) a copper-cobalt-nickel mine within mallee woodland in the Warm semi-arid region; (E) a nickel mine within open shrubland and grassland in the Hot desert region; and (F, G) a bauxite mine and another undisclosed commodity mine both within jarrah forest in the Warm-summer mediterranean region. Inset: location of Western Australia within Australia.

## 2.1 Sampling design

At each of the seven mine sites, three plots were established in undisturbed native vegetation (hereafter referred to as 'reference plots') and four in topsoil stockpiles (i.e., salvaged natural topsoil stockpiled for later re-spreading) of varying age (ages range from 3 months to 29 years) in summer 2018/2019. Five 5 m × 5 m subplots were established at each reference and stockpile plot, with five replicate soil samples collected at random from the top 0–20 cm soil layer in each subplot using a 5 cm diameter soil auger. At the youngest stockpile at each mine, five additional samples were taken from the 50–70 cm depth. The five soil samples were bulked and homogenised to produce one composite sample per subplot. Sterile nitrile gloves were worn when collecting soils, sampling equipment was sterilised with a bleach solution between all samples, and gloves were changed between plots to prevent cross-contamination. Subsamples were taken from each composite sample and stored at -20°C until DNA could be extracted (see below), while the remainder of the composites were dried in an oven at 40°C for 48 hours before being sieved (2 mm gauge). Subsamples of the dried, composite soil were reserved for chemical analysis, assessment of carbon



dioxide production, and analyses of soil microbial community composition. In total 280 soil samples were collected from the seven mine sites (i.e. at each mine there were three reference plots, four stockpile plots, and one young stockpile plot. Each plot consisted of 5 subplots and one composite sample was collected from each subplot).

## 2.2 Conventional soil analyses

To provide comparison data for the 29 soil chemical, physical and biological soil properties to be assessed using spectroscopic methods, soil organic carbon content (Walkley and Black, 1934), potassium and phosphorus content (Colwell, 1965), pH (in a 1:5 soil to 0.01M $CaCl_2$ and $H_2O$), electrical conductivity, sulphur, ammonium nitrogen, nitrate nitrogen, boron, trace elements (DTPA; copper, zinc, manganese, iron), and exchangeable cations (calcium, magnesium, sodium, potassium, aluminium) were determined for all composite samples using analytical chemistry methods *sensu* Rayment and Lyons (2010). Soil particle size
(percentage sand, silt, clay), and bulk density were also assessed for each soil sample (Rai et al., 2017). Microbial activity of each soil sample was assessed using the Solvita 1-Day $CO_2$ Test (Haney et al. (2008); Munoz-Rojas et al. (2016)). The test was conducted as per the manufacturer's instructions (2019 SOP; Woods End Laboratories Inc., 2018-2019); briefly, 30 $cm^3$ of dried soil was re-wet with 9 mL of water and placed in a sealed container with a Solvita $CO_2$ probe for 24 hours, after which a digital colour reader (for use with the $CO_2$ test; Solvita, Woods End Laboratories, USA) was used to measure the volume of
carbon dioxide produced.

Soil microbial community composition was determined by extracting DNA from 250 mg of homogenised soil samples (DNeasy PowerSoil HTP 96 Kit; Qiagen, Germany). DNA quality and level of inhibition were checked through quantitative PCR (qPCR), with negative PCR controls included (Murray et al., 2015). Primer sets used targeted the V4 16S rRNA region for Bacteria (Turner et al., 1999; Caporaso et al., 2011) and the Internal Transcribed Spacer 2 for fungi (Ihrmark et al., 2012; White
et al., 1990). Single fusion sequencing (see Supplementary Information 'DNA Sequencing and Bioinformatics Methodology') was performed at the Trace and Environmental DNA laboratory, Curtin University (Bentley, WA) on the MiSeq platform (Illumina, USA) as per (van der Heyde et al., 2021).

Sequences were de-multiplexed, quality filtered, error rates estimated, and de-replicated to leave Amplicon Sequence Variants. Taxonomy was assigned based on reference databases (SILVA (Quast et al., 2013) for bacteria and UNITE8.2 (Nilsson
et al., 2019) for fungi). Alpha diversity and richness of fungal and bacterial taxa were calculated from sequence data using the Shannon Index (Wagner et al., 2018).

The statistical distributions of electrical conductivity, organic C, Total N, K (Colwell), B, S, Cu, Fe, Mn, exchangeable Mg and Na, Ammonium-N, Nitrate-N, and $CO_2$ exhibited strong positive skew, and were thus log transformed to approximate normal distribution prior to use in spectroscopic modelling.

## 2.3 Soil spectroscopy

For spectroscopic analysis, the $\leq$ 2 mm-sieved composite subsamples were each gently mixed and then placed in a petri-dish. Diffuse reflectance spectra were recorded using four miniaturised spectrometers with different spectral ranges, resolutions, dimensions and costs (Table 2). We measured the soils following protocols described in (Viscarra Rossel et al., 2016, Appendix





B). The spectrometers were switched on an hour before measurements and the control software of each instrument was set up
to record (and average) 30 readings per soil sample measurement and 50 readings per calibration measurement. Calibration was
performed with a Halon white reference (Spectralon®) and dark internal reference. The spectrometers were recalibrated every
ten measures. We performed the spectroscopic measurements of the soil samples in two separate rounds (i.e. in replicate), by
the same analyst and under the same laboratory conditions.

**Table 2.** Spectral range, resolution, price, weight, and dimensions of miniaturised and portable spectrometers used in this study.

| Label | Device name | Manufacturer (location) | Spectral range (nm) | Resolution (nm) | Price (AUD) | Weight (g) | Dimensions (mm) |
|---|---|---|---|---|---|---|---|
| $A_{350-830}$ | STS-VIS | Ocean Insight, (Orlando, Florida) | 350–830 | 1.5, 3, 6, 12 | 4800 | 60 | $40 \times 42 \times 24$ |
| $B_{1750-2150}$ | NIRONE Sensor S | Spectral Engines, (Steinbach, Germany) | 1750–2150 | 16–22 | 5,120 | 15 | $25 \times 25 \times 18$ |
| $C_{1450-2450}$ | trinamiX | trinamiX, (Ludwigshafen, Germany) | 1450–2450 | 15–25 | 12,300 | 560 | $152 \times 84 \times 52$ |
| $D_{1300-2600}$ | NeoSpectra | Si-Ware Systems, (El Nozha, Cairo) | 1300–2600 | 16 | 5,000 | 17 | $32 \times 32 \times 22$ |
| $E_{350-2500}$ | SR-3500 | Spectral Evolution, (Haverhill, Massachusetts) | 350–2500 | 2.8, 6, 8 | 70,000 | 3800 | $216 \times 279 \times 89$ |

The reflectance (R) spectra of the soil samples recorded with each instrument were transformed to apparent absorbance
using $\log_{10}(1/R)$, and interpolated to 10 nm intervals to attain a consistent wavelength interval. Since some of the miniaturised
spectrometers measured narrow and specific spectral ranges, we combined spectrometers $A_{350-830}$ with $B_{1750-2150}$, $A_{350-830}$
with $C_{1450-2450}$, and $A_{350-830}$ with $D_{1300-2600}$, to cover wider spectral ranges.

## 2.4   Spectroscopic modelling

To assess the spectroscopic modelling with different statistical and machine learning algorithms, as well as the accuracy of the
spectrometers estimates and their repeatability, the modelling was undertaken using aggregated subplot data, which gave 56
data (from the 21 reference, 35 stockpile plots) for modelling. Modelling was performed using the average spectra from the two
replicates, except when assessing repeatability of the spectrometers. We also modelled the soil properties using all 280 data
(from the 105 reference, 175 stockpile subplots) to verify the spectroscopic modelling and the accuracy of the spectrometers
estimates. Below, we describe the modelling in some detail.

### 2.4.1   Assessment of the spectroscopic modelling algorithms with data from plots

We used seven statistical and machine learning methods to model the soil properties with the spectra from each instrument
and their combinations. These were partial least square regression (PLSR) (Wold et al., 2001), random forest (RF) (Breiman,
2001), support vector machines (SVM) (Vapnik, 1999), Cubist (Quinlan et al., 1992), extreme gradient boosting (XGBoost)
(Chen et al., 2015), and Gaussian process regression with linear (GPRL) and polynomial (GPRP) kernels (Rasmussen, 2003).
Viscarra Rossel and Behrens (2010) describe these algorithms and their implementation in spectroscopic soil modelling. The
models were developed using 10-fold cross-validation and their hyperparameters were optimised by minimising the root mean
squared error (RMSE) with either grid search or using the Differential Evolution optimisation (Price et al., 2006), implemented
in the R library DEOPTIM (Mullen et al., 2011). The optimal number of factors to use in the PLSR was determined using a
grid search, implemented in the R library PLS (Wehrens and Mevik, 2007). We implemented the SVM using a Gaussian radial



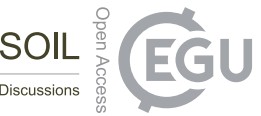

basis function in the R library KERNLAB (Karatzoglou et al., 2004). Its hyperparameters C, which describes the cost or penalty on training accuracy and behaves as a regularization parameter in the SVM, and $\gamma$, which defines the influence of training samples, were optimized using DEOPTIM. RF was implemented using the R library RANDOMFOREST (Liaw et al., 2002) and the hyperparameter mtry, which is the number of predictors randomly sampled as candidates at each decision tree split, was optimised with DEOPTIM. We implemented GPR using the R library KERNLAB and tested both linear and polynomial kernels.

The linear method did not need optimising; however, for the polynomial kernel, we used DEOPTIM to optimise the degree of the polynomial and scale hyperparameter. The optimisation of hyperparameters in XGBoost, implemented using the R library XGBOOST was also performed using DEOPTIM. The hyperparameters optimised were number of iterations (nrounds), the learning rate (eta), maximum tree depth (max_depth), the regularisation parameter, which controls overfitting ($\gamma$), the number of predictors supplied to each tree (colsample_bytree), the minimum number of instances required in a child node

(min_child_weight), and the number of samples (observations) supplied to a tree (subsample). Viscarra Rossel and Webster (2012) described the implementation of Cubist in spectroscopic modelling. Its hyperparameters, the number of committees (c) and neighbours (N), were optimised with DEOPTIM.

For each soil property, we calculated the mean, maximum, and minimum concordance correlation coefficient (Lin, 1989), $\rho_c$, of each algorithm and selected only the most accurate and consistent ones to compare the performance of the spectrometers and

their combinations. We removed algorithms with an average $\rho_c$ below 0.65 as they performed poorly with most spectrometers and soil properties. This improved the interpretability of our results and removed redundant models that were unsuitable for modelling the data.

### 2.4.2 Assessment of the spectroscopic modelling with the different spectrometers

To compare the performance of the different spectrometers we modelled the 29 soil properties using the five spectrometers

and the three spectrometer combinations. We first selected the algorithms yielding the most consistent estimates (see above) for the soil properties and then calculated the average, minimum and maximum $\rho_c$ across the best algorithms to assess the spectrometer's and the combinations' accuracy for each property.

The $\rho_c$ is a unit invariant coefficient that measures the difference between the measured and estimated values and their deviation from a 45-degree line of perfect agreement, evaluating both precision and bias (Lin, 1989). We used $\rho_c < 0.65$ to

denote poor agreement between the measured values of the properties and the estimates, $0.65 \leq \rho_c < 0.8$ to denote moderate agreement, $0.8 \leq \rho_c < 0.9$ for substantial agreement, and $\rho_c \geq 0.9$ for near-perfect agreement. A value of 1 denotes perfect agreement.

To more explicitly quantify the estimation error with the spectrometer or spectrometer combinations that produced the best predictions (i.e. the largest $\rho_c$), we measured the root mean squared error (RMSE), the mean error (ME) and the standard

deviation of the error (SDE), which represent the inaccuracy, bias and imprecision of the estimates, respectively. We note that the inaccuracy (RMSE) embraces both the bias (ME) and the imprecision (SDE) of the analysis so that $RMSE^2 = ME^2 + SDE^2$.





### 2.4.3   Repeatability of the spectrometers

We assessed the repeatability of the spectrometer measurement by calculating the ratio of the difference between the replicates to the mean of the replicates, defined as:

$$\%\text{Difference} = \frac{|\text{Rep}_a - \text{Rep}_b|}{(\text{Rep}_a + \text{Rep}_b)/2} \times 100 \tag{1}$$

where $\text{Rep}_a$ and $\text{Rep}_b$ represent the two spectroscopic replicate measurements. Repeatability values (% Difference) closer to zero are considered to represent more repeatable spectroscopic measurements. To assess the impact a spectrometer's repeatability had on the spectroscopic modelling, we also modelled soil properties with each replicate ($\text{Rep}_a$ and $\text{Rep}_b$) and calculated the absolute difference in $\rho_c$, $\Delta\rho_c$, of the estimates. We performed the modelling using the best algorithms (see above) and report the mean, maximum, and minimum $\Delta\rho_c$ for each spectrometer and combination.

### 2.4.4   Assessment of the spectroscopic modelling with data from subplots

To further verify spectroscopic modelling from averaged plot data, we also evaluated the performance of the spectrometers using data from subplots. When modelling the 280 subplot data, we used only the algorithm that performed best in modelling the data from the plots. As the subplots originated from within the reference and stockpiles plots, we performed the modelling and assessment using 10-fold-plot-out cross-validation to prevent the risk of overfitting because soil properties and spectra from a reference or stockpile plot can be correlated.

## 2.5   Overall assessment of the spectrometers

To assess the overall performance of each spectrometer and combination as a function of their accuracy and repeatability, we derived the index $e$:

$$e = \frac{\sum_{i=1}^{N} \overline{\rho_c^i}}{N} + \frac{\sum_{i=1}^{N}(1 - (^{max}\rho_c^i - ^{min}\rho_c^i))}{N} + \frac{\sum_{i=1}^{N}(1 - \overline{\Delta\rho_c^i})}{N} + \frac{\sum_{i=1}^{N}(1 - (^{max}\Delta\rho_c^i - ^{min}\Delta\rho_c^i))}{N} \tag{2}$$

where, $e$ is the overall performance index, $N$ is the number of soil properties, $\overline{\rho_c^i}$, $^{max}\rho_c^i$ and $^{min}\rho_c^i$ are the mean, maximum and minimum $\rho_c$ for the $i$th property from the best models, $\overline{\Delta\rho_c^i}$, $^{max}\Delta\rho_c^i$ and $^{min}\Delta\rho_c^i$ are the mean, maximum and minimum difference in $\rho_c$ from the replicate measurements for the $i$th property from the best algorithms. The first two terms in Equation 2 assesses the overall accuracy and stability of a spectrometers or combinations when the calibrations are performed using different algorithms. The third and fourth terms assess the effect of the spectrometers repeatability on the spectroscopic modelling and its stability. All terms range from 0 to 1, and a higher value means better performance.

## 3   Results

The soil samples varied markedly in their physical, chemical, and biological properties, providing a wide range of values for spectroscopic modelling. The soils represent a range from healthy, reference sites to degraded stockpiles, as evidenced by the



wide range of their properties. For example, clay content ranged from 4–66%, organic C from 0.19–4.3%, pH 5.12 to 9.1 and microbial activity ($CO_2$ flux) from 5–140 mg $L^{-1}$ (Table 3).

**Table 3.** Summary of the physical (n=5), chemical (n=19), and biological (n=5) properties (mean, s.d., minimum, maximum, median, and 1st and 3rd quartiles) of reference soil and stockpiled topsoil sampled from seven mine sites in Western Australia, as determined by conventional analytical methods.

| Soil property | Unit | Mean | s.d. | Min. | $1^{st}$ Quart. | Median | $3^{rd}$ Quart. | Max. |
|---|---|---|---|---|---|---|---|---|
| **Physical properties** | | | | | | | | |
| Sand | % | 51.44 | 26.38 | 18.00 | 32.00 | 32.80 | 90.00 | 90.00 |
| Silt | % | 24.10 | 12.22 | 6.00 | 6.00 | 30.60 | 34.00 | 37.60 |
| Clay | % | 24.46 | 17.12 | 4.00 | 4.00 | 32.60 | 34.00 | 66.00 |
| Bulk density | g cm$^{-3}$ | 1.36 | 0.15 | 0.81 | 1.30 | 1.360 | 1.47 | 1.58 |
| Electrical conductivity | dS m$^{-1}$ | 0.19 | 0.49 | 0.0128 | 0.027 | 0.044 | 0.15 | 2.87 |
| **Biological properties** | | | | | | | | |
| $CO_2$ production | mg L$^{-1}$ | 30.41 | 29.22 | 5.20 | 11.60 | 21.16 | 38.36 | 140.06 |
| Fungal richness | - | 52.26 | 39.87 | 8.00 | 26.94 | 45.10 | 63.19 | 176.00 |
| Fungal diversity | - | 2.82 | 0.65 | 1.46 | 2.33 | 2.81 | 3.42 | 4.13 |
| Bacterial richness | - | 656.64 | 249.81 | 171.75 | 464.62 | 641.50 | 798.73 | 1439.00 |
| Bacterial diversity | - | 5.58 | 0.60 | 3.98 | 5.31 | 5.65 | 6.00 | 6.46 |
| **Chemical properties** | | | | | | | | |
| pH$_{Ca}$ (0.01M CaCl$_2$) | - | 5.74 | 1.20 | 3.88 | 4.92 | 5.45 | 6.21 | 8.16 |
| pH$_W$ | - | 6.68 | 1.06 | 5.12 | 5.90 | 6.36 | 7.11 | 9.10 |
| Organic C | % | 1.21 | 1.07 | 0.19 | 0.43 | 0.86 | 1.83 | 4.30 |
| Total N | mg kg$^{-1}$ | 14.52 | 23.50 | 2.00 | 4.71 | 7.280 | 13.30 | 155.6 |
| P (Colwell) | mg kg$^{-1}$ | 4.53 | 2.62 | 1.00 | 3.15 | 4.20 | 5.60 | 13.2 |
| K (Colwell) | mg kg$^{-1}$ | 191.71 | 145.15 | 15.00 | 41.75 | 194.60 | 297.90 | 471.40 |
| B | mg kg$^{-1}$ | 0.73 | 1.04 | 0.10 | 0.24 | 0.36 | 0.58 | 4.47 |
| S | mg kg$^{-1}$ | 60.13 | 225.00 | 0.70 | 2.98 | 5.95 | 19.44 | 1467 |
| Cu | mg kg$^{-1}$ | 0.84 | 0.72 | 0.09 | 0.19 | 0.79 | 1.36 | 2.96 |
| Fe | mg kg$^{-1}$ | 18.24 | 16.89 | 2.96 | 6.82 | 11.02 | 28.21 | 79.20 |
| Mn | mg kg$^{-1}$ | 13.94 | 15.64 | 0.62 | 4.49 | 10.14 | 17.62 | 88.88 |
| Zn | mg kg$^{-1}$ | 0.56 | 0.35 | 0.07 | 0.29 | 0.40 | 0.72 | 1.47 |
| Exchangeable Ca | meq 100 g$^{-1}$ | 6.09 | 5.67 | 0.64 | 2.12 | 3.57 | 8.29 | 20.17 |
| Exchangeable Mg | meq 100 g$^{-1}$ | 1.91 | 1.81 | 0.15 | 0.74 | 1.15 | 2.55 | 6.38 |
| Exchangeable Na | meq 100 g$^{-1}$ | 0.65 | 1.53 | 0.024 | 0.060 | 0.097 | 0.43 | 8.30 |
| Exchangeable K | mg 100 kg$^{-1}$ | 0.38 | 0.31 | 0.01 | 0.089 | 0.33 | 0.56 | 1.02 |
| Exchangeable Al | meq 100 g$^{-1}$ | 0.12 | 0.14 | 0.018 | 0.035 | 0.060 | 0.12 | 0.57 |
| Ammonium-N | mg kg$^{-1}$ | 3.39 | 3.27 | 1.00 | 1.40 | 2.30 | 4.00 | 19.8 |
| Nitrate-N | mg kg$^{-1}$ | 11.13 | 21.66 | 1.00 | 1.20 | 4.60 | 10.95 | 143.8 |





The reflectance spectra from the different spectrometers and their combinations show recognisable features that are characteristic of soil spectra. For example, the feature near 1900 nm (Fig. 2), which is due to combination and overtone vibrations of molecular water contained within soil minerals (Viscarra Rossel and Behrens, 2010).

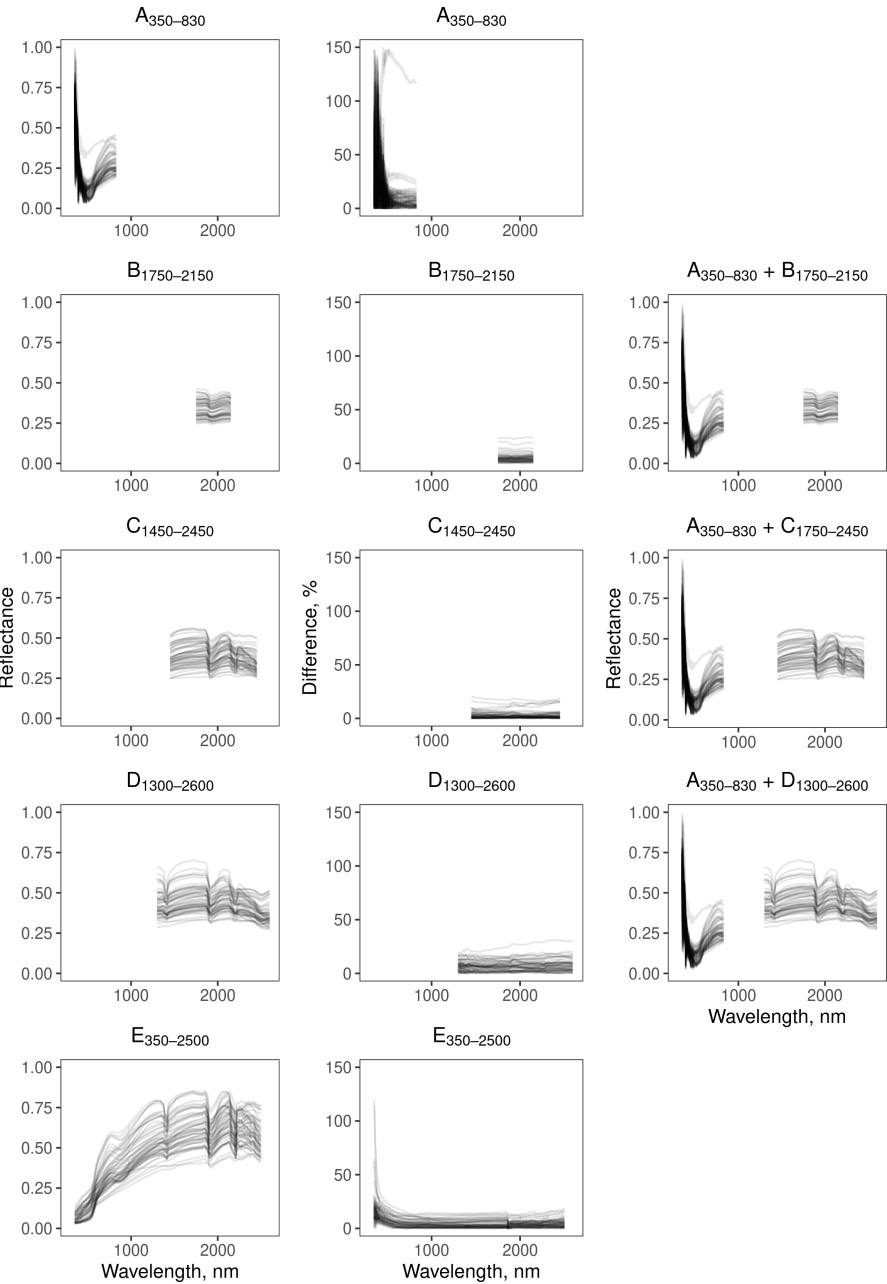

**Figure 2.** Mean and difference of the spectral replicates. Column one: mean reflectance spectra of the replicates; Column two: difference between the spectral replicates calculated using Equation (1); Column three: combined reflectance spectra.





The $E_{350-2500}$ was more repeatable in the range between 600–2500 nm with less than 18.7% difference between the replicates. At wavelengths smaller than 600 nm, however, the difference between the replicates was large (Fig. 2). The repeatability of the $B_{1750-2150}$ (< 24.3%), $C_{1450-2450}$ (< 20.6%) and $D_{1300-2600}$ (< 30.9%) spectrometers was similar, with $B_{1750-2150}$ and $C_{1450-2450}$ being slightly more repeatable than $D_{1300-2600}$. In contrast, replicate spectra from $A_{350-830}$ were more imprecise in the 350–500 nm region (Fig. 2).

## 3.1 Assessment of the spectroscopic modelling algorithms

PLSR, GPRP, Cubist and GPRL consistently produced more accurate estimates (mean $\rho_c$ of 0.74, 0.74, 0.71, and 0.69 respectively) of the soil physical, chemical and biological properties than SVM, RF and XGBoost (mean $\rho_c$ of 0.51, 0.54, and 0.58 respectively), with less variability (narrower minimum, maximum intervals) between spectrometers (Fig. 3).

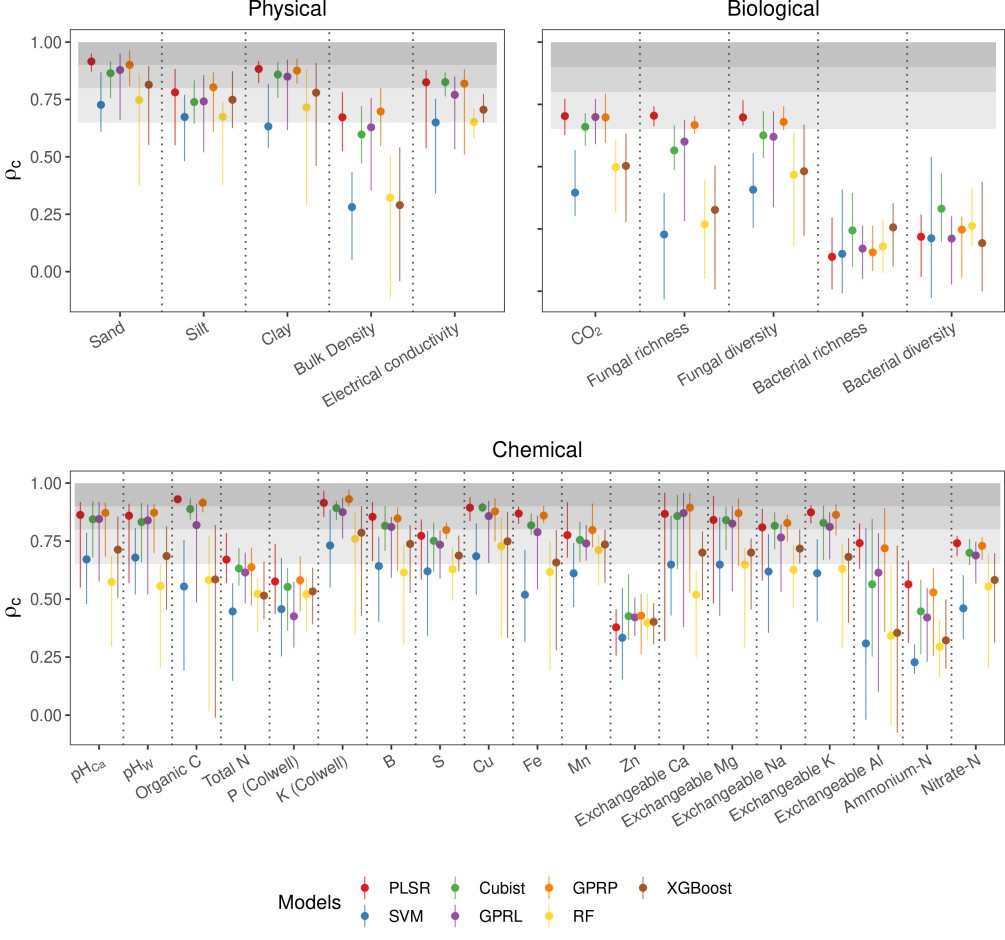

**Figure 3.** Assessment of the accuracy from different algorithms. The mean $\rho_c$ (points) and range (lines) extend the minimum and maximum values of $\rho_c$ from the five spectrometers and three combinations. The three shaded grey bands indicate the accuracy level, which we classified as poor ($\rho_c < 0.65$), moderate ($0.65 \leq \rho_c < 0.8$), substantial ($0.8 \leq \rho_c < 0.9$), and near perfect ($0.9 \leq \rho_c \leq 1.0$).





Of the soil physical properties, the algorithms most poorly estimated bulk density ($\rho_c < 0.65$) (Fig. 3). All of the algorithms
performed well for the other soil physical properties ($\rho_c \geq 0.65$). The accuracy of the estimates of the biological properties
were varied, $CO_2$ flux with PLSR, GPRP, Cubist and GPRL produced $\rho_c$ values between 0.65–0.8 (Fig. 3). Estimates of fungal
diversity and fungal richness with these algorithms varied and only PLSR and GPRP produced mean values of $\rho_c$ in the range
0.65–0.8. Although the Cubist estimates were markedly better, all algorithms poorly estimated bacterial richness and diversity
($\rho_c < 0.65$) (Fig. 3). The algorithms poorly estimated the chemical properties P, Zn, and Ammonium-N ($\rho_c < 0.65$). The
chemical properties, $pH_{Ca}$, $pH_W$, organic C, K, Cu, exchangeable Ca, were predicted with substantial accuracy ($\rho_c \geq 0.8$) by
PLSR, Cubist, GPRP and GPRL. The variability in the estimates of the exchangeable Ca, Mg and Al, was large compared to
other chemical properties, arising from the performance of the different spectrometers (Fig. 3).

### 3.2   Assessment of the spectroscopic modelling with the spectrometers

In this section, we assess the different spectrometers and combinations considering only the algorithms that performed best:
PLSR, GPRP, Cubist and GPRL (see above and Fig. 3). Combining both visible and NIR generally produced more accurate
estimates of the soil physical, biological and chemical properties compared to the visible or NIR spectrometers (Fig. 4).
Generally, the visible $A_{350–830}$ was accurate for soil texture (sand, silt, and clay) and least accurate for chemical properties. The
NIR spectrometers predicted most of the soil physical, chemical, and biological properties with moderate or greater accuracy
($\rho_c \geq 0.65$). The $B_{1750–2150}$ spectrometer, with the narrowest spectral range produced the least accurate estimates of the soil
physical and biological properties (Fig. 5).

The $A_{350–830}$ spectrometer produced the best estimates of silt content, and its estimates of sand and clay were comparably
accurate to those made using instruments that cover the NIR and vis–NIR ranges (Fig. 4). Estimates of sand and silt content
using only the NIR range, with the $B_{1750–2150}$, $C_{1450–2450}$ and $D_{1300–2600}$ spectrometers were the least accurate. Estimates of
soil electrical conductivity with the visible, NIR and vis–NIR ranges were similar ($0.8 \leq \rho_c < 0.9$, Fig. 4). The spectrometers
that combined the visible and NIR ranges tended to estimate better the soil biological properties. For instance, the $A_{350–830}$ +
$D_{1300–2600}$ and $E_{350–2500}$ produced the most accurate estimate of $CO_2$; $A_{350–830}$ + $B_{1750–2150}$ and $E_{350–2500}$ produced better
estimates of fungal richness and fungal diversity respectively (Fig. 4). Of the soil chemical properties that were estimated with
at least moderate accuracy ($\rho_c \geq 0.65$), the $C_{1450–2450}$, $D_{1300–2600}$, $A_{350–830}$ + $B_{1750–2150}$, $A_{350–830}$ + $C_{1450–2450}$, $A_{350–830}$
+ $D_{1300–2600}$ and $E_{350–2500}$ produced estimates with comparable accuracy (Fig. 4), whereas the $A_{350–830}$ produced the least
accurate results.





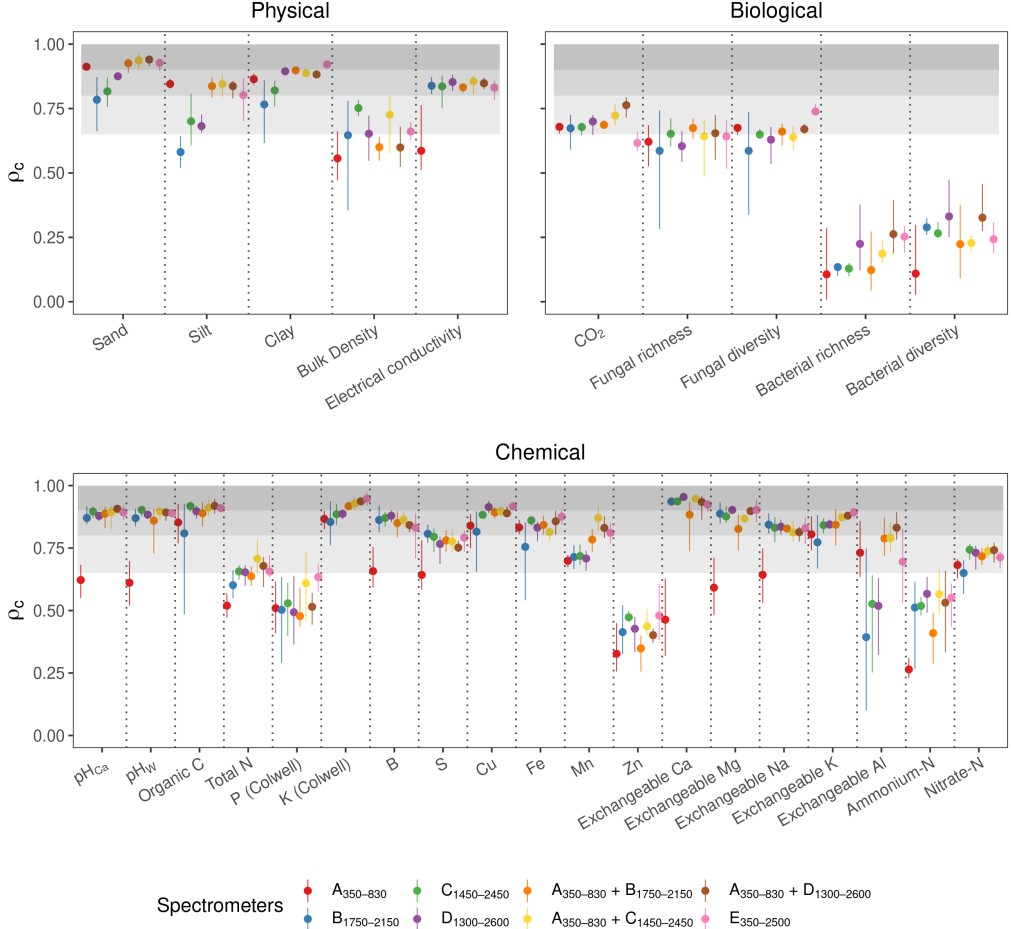

**Figure 4.** Accuracy of the spectrometers and combinations. The discs show the mean $\rho_c$ and the range lines extend the minimum and maximum values of $\rho_c$ from the best four algorithms (PLSR, Cubist, GPRL, and GPRP). The grey bands indicate the level of accuracy (moderate, substantial, and near perfect—see caption of Figure 3).

### 3.3 Effect of repeatability on the spectroscopic modelling

For all of the soil physical, chemical and biological properties, the $E_{350–2500}$ full range vis–NIR portable spectrometer produced estimates that were generally the most precise in terms of their repeatability (Fig. 5). However, the repeatability of the estimates from the miniaturised NIR ($B_{1750–2150}$, $C_{1450–2450}$, and $D_{1300–2600}$) and combined vis–NIR ($A_{350–830}$ + $B_{1750–2150}$, $A_{350830}$ +

$C_{1450–2450}$ and $A_{350–830}$ + $D_{1300–2600}$) spectrometers were comparable.

Due to the poor repeatability of the $A_{350–830}$ measurements (see Fig. 2), the estimates of the soil properties with the spectra from this instrument were the most uncertain, particularly the chemical properties (Fig. 5). For a number of properties that were not well estimated with the $A_{350–830}$ instrument (e.g. bulk density, S, and Exchangeable Na and Al), combining it with a NIR spectrometer affected the precision of the spectrometer combinations. For the soil properties that could not be well estimated




(bacterial richness and diversity, P, Zn and Ammonium-N with $\rho_c < 0.65$), the precision of the estimates with all spectrometers was poor.

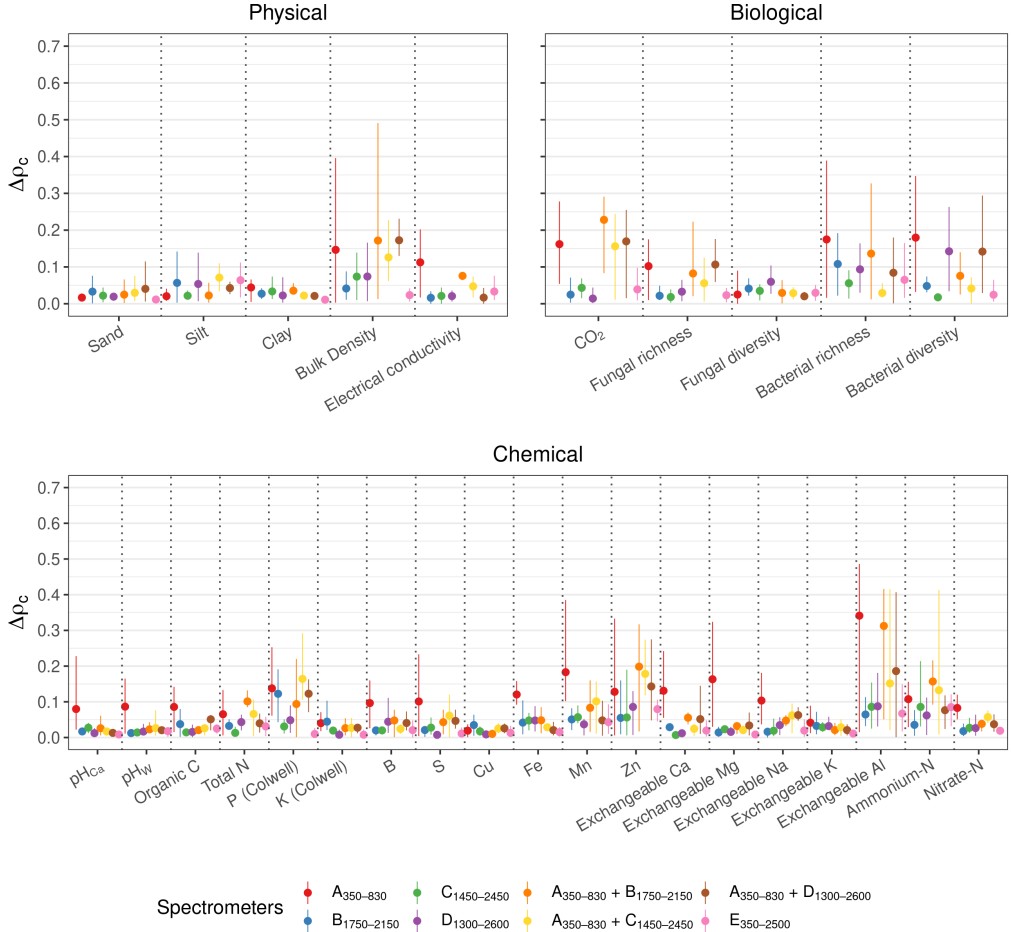

**Figure 5.** Effect of the repeatability of the spectrometers on the spectroscopic modelling. $\Delta\rho_c$ represents the difference in $\rho_c$ between the modelling (with PSLR, Cubist, GPRL, and GPRP) of the replicates. The discs show the mean difference and the range lines extend the minimum and maximum of the difference. A small $\Delta\rho_c$ indicates good repeatability.

### 3.4 Overall assessment of the spectrometers

Spectrometers that cover the visible and NIR ranges were the most accurate and stable (Table 4). The $A_{350–830}$ + $C_{1750–2450}$ and $A_{350–830}$ + $D_{1300–2600}$ spectrometers had the same accuracy and stability as the full-range $E_{350–2500}$. Spectrometers,

$C_{1450–2450}$ and $E_{350–2500}$ were the most repeatable, while $A_{350–830}$ was the least repeatable (Table 4). The precision of $A_{350–830}$ + $C_{1450–2450}$ and $A_{350–830}$ + $D_{1300–2600}$ was affected by the poor repeatability of $A_{350–830}$. Overall, $E_{350–2500}$ had the highest $e$ score, followed by $C_{1450–2450}$, $D_{1300–2600}$, $A_{350–830}$ + $C_{1450–2450}$, and $A_{350–830}$ + $D_{1300–2600}$.





**Table 4.** Overall assessment of the spectrometers and combinations. Columns show the accuracy and stability (represented by the range of the accuracy) of the spectrometers when modelling with the different algorithms (PSLR, Cubist, GPRL, and GPRP) as well as the effect of the spectrometers repeatability on the modelling and its stability. Respectively, they corresponds to the first, second, third and forth term in Equation 2.

| Spectrometer | Accuracy | Accuracy range | Repeatability | Repeatability range | $e$ |
|---|---|---|---|---|---|
| $A_{350-830}$ | 0.62 | 0.85 | 0.89 | 0.82 | 3.18 |
| $B_{1750-2150}$ | 0.68 | 0.79 | 0.96 | 0.94 | 3.37 |
| $C_{1450-2450}$ | 0.72 | 0.91 | 0.97 | 0.95 | 3.55 |
| $D_{1300-2600}$ | 0.72 | 0.90 | 0.96 | 0.93 | 3.51 |
| $A_{350-830} + B_{1750-2150}$ | 0.72 | 0.89 | 0.92 | 0.89 | 3.42 |
| $A_{350-830} + C_{1450-2450}$ | 0.75 | 0.91 | 0.94 | 0.90 | 3.50 |
| $A_{350-830} + D_{1300-2600}$ | 0.75 | 0.91 | 0.94 | 0.91 | 3.51 |
| $E_{350-2500}$ | 0.75 | 0.91 | 0.97 | 0.95 | 3.58 |

For each soil property we derived the spectrometer performance score, $e$, for all the miniaturised spectrometers and combinations, and listed the statistics of the spectrometer/combination with the largest $e$ in Table 5. The miniaturised spectrometers and combinations estimated 24 of the 29 soil properties with moderate or better accuracy ($\rho_c \geq 0.65$), except for P, Zn, Ammonium-N, and bacterial properties. For all the properties, the imprecision was larger than the bias.

The visible range spectrometer $A_{350-830}$ had the highest $e$ for sand and silt with perfect ($\rho_c \geq 0.90$) and substantial ($0.80 \leq \rho_c \leq 0.90$) accuracy respectively (Table 5). The individual NIR spectrometers performed well on many chemical and some of the biological properties. The $C_{1450-2450}$ spectrometer alone predicted one soil physical property, six chemical properties and one biological property with moderate or better accuracy ($\rho_c \geq 0.65$). Some properties had the best estimates when the visible spectrometer was combined with an NIR one. The combination $A_{350-830} + D_{1300-2600}$ covered most of these properties.





**Table 5.** Statistics from miniaturised spectrometers and combinations with the highest $e$ score for individual soil properties.

| Soil property | Spectrometer/combination | $e$ | $\rho_c$ | RMSE | ME | SDE |
|---|---|---|---|---|---|---|
| **Physical properties** | | | | | | |
| Sand | $A_{350-830}$ | 3.86 | 0.91 | 10.64 | -0.40 | 10.63 |
| Silt | $A_{350-830}$ | 3.75 | 0.85 | 6.43 | -0.023 | 6.43 |
| Clay | $A_{350-830} + D_{1300-2600}$ | 3.83 | 0.88 | 8.09 | -0.10 | 8.09 |
| Bulk density | $C_{1450-2450}$ | 3.49 | 0.75 | 0.093 | 0.0038 | 0.093 |
| $\log_{10}$(Electrical conductivity) | $A_{350-830} + D_{1300-2600}$ | 3.74 | 0.85 | 0.29 | 0.0093 | 0.28 |
| **Chemical properties** | | | | | | |
| $pH_{Ca}$ | $A_{350-830} + D_{1300-2600}$ | 3.85 | 0.91 | 0.49 | -0.0073 | 0.49 |
| $pH_W$ | $C_{1450-2450}$ | 3.84 | 0.90 | 0.45 | -0.0085 | 0.45 |
| $\log_{10}$(OC) | $C_{1450-2450}$ | 3.85 | 0.92 | 0.14 | 0.0060 | 0.14 |
| $\log_{10}$(Total N) | $C_{1450-2450}$ | 3.56 | 0.66 | 0.29 | 0.017 | 0.29 |
| P | $C_{1450-2450}$ | 3.24 | 0.53 | 2.20 | 0.051 | 2.20 |
| $\log_{10}$(K) | $A_{350-830} + D_{1300-2600}$ | 3.86 | 0.94 | 0.17 | -0.0063 | 0.17 |
| $\log_{10}$(B) | $C_{1450-2450}$ | 3.78 | 0.87 | 0.20 | -0.0035 | 0.20 |
| $\log_{10}$(S) | $B_{1750-2150}$ | 3.69 | 0.81 | 0.41 | -0.0038 | 0.41 |
| $\log_{10}$(Cu) | $D_{1300-2600}$ | 3.84 | 0.91 | 0.18 | 0.0040 | 0.18 |
| $\log_{10}$(Fe) | $C_{1450-2450}$ | 3.73 | 0.86 | 0.17 | -0.0030 | 0.17 |
| $\log_{10}$(Mn) | $A_{350-830} + D_{1300-2600}$ | 3.60 | 0.83 | 0.27 | 0.0023 | 0.27 |
| Zn | $C_{1450-2450}$ | 3.19 | 0.47 | 0.30 | 0.0063 | 0.30 |
| Exchangeable Ca | $D_{1300-2600}$ | 3.93 | 0.95 | 1.69 | 0.019 | 1.69 |
| $\log_{10}$(Exchangeable Mg) | $D_{1300-2600}$ | 3.85 | 0.90 | 0.18 | 0.0043 | 0.18 |
| $\log_{10}$(Exchangeable Na) | $A_{350-830} + B_{1750-2150}$ | 3.74 | 0.83 | 0.36 | 0.015 | 0.36 |
| Exchangeable K | $A_{350-830} + D_{1300-2600}$ | 3.81 | 0.88 | 0.15 | -0.0013 | 0.15 |
| Exchangeable Al | $A_{350-830} + D_{1300-2600}$ | 3.13 | 0.83 | 0.078 | -0.0033 | 0.078 |
| $\log_{10}$(Ammonium-N) | $D_{1300-2600}$ | 3.26 | 0.57 | 0.25 | 0.0053 | 0.25 |
| $\log_{10}$(Nitrate-N) | $C_{1450-2450}$ | 3.61 | 0.74 | 0.36 | 0.029 | 0.36 |
| **Biological properties** | | | | | | |
| $\log_{10}(CO_2)$ | $D_{1300-2600}$ | 3.57 | 0.70 | 0.26 | 0.0065 | 0.26 |
| Fungal richness | $C_{1450-2450}$ | 3.49 | 0.65 | 27.35 | 1.14 | 27.33 |
| Fungal diversity | $A_{350-830} + D_{1300-2600}$ | 3.59 | 0.67 | 0.44 | -0.0088 | 0.44 |
| Bacterial richness | $A_{350-830} + C_{1450-2450}$ | 3.03 | 0.19 | 247.13 | -1.26 | 247.13 |
| Bacterial diversity | $C_{1450-2450}$ | 3.18 | 0.27 | 0.56 | -0.014 | 0.56 |





**Spectroscopic modelling with PLSR and 10-fold-plot-out cross validation**

Compared to the other algorithms tested, overall, PLSR produced more accurate estimates of the soil properties (see Fig. 3), so we used it for modelling the 280 data from the subplots (see Methods section Assessment of the spectroscopic modelling with
data from subplots). Since the $A_{350-830}$ and $B_{1750-2150}$ spectrometers generally produced the least accurate estimates (Fig. 4) and had the smallest $e$ score (Table 4), we did not use them in this modelling. The 10-fold-plot-out cross-validations of the soil properties from subplots (Fig. 6) were similar or slightly more conservative compared to those of the 10-fold cross-validation of the data from plots (Fig. 4).

**Figure 6.** Ten-fold-plot-out cross-validation using PLSR.





The accuracy of the estimates with the combinations $A_{350-830}$ + $C_{1450-2450}$ and $A_{350-830}$ + $D_{1300-2600}$ spectrometers were

similar to the full range portable $E_{350-2500}$ spectrometer and better than the combined $A_{350-830}$ + $B_{1750-2150}$ spectrometer (Fig. 6). The $C_{1450-2450}$ and $D_{1300-2600}$ have similar accuracy and were less accurate than $A_{350-830}$ + $C_{1450-2450}$ and $A_{350-830}$ + $D_{1300-2600}$ respectively.

## 4  Discussion

Mining is vital to economic development in many countries. In Australia, for example, mining and energy exports are forecast
to be worth around \$AU288 billion in 2020/21, and over \$184 billion of this from WA alone (Department of Industry, Science, Energy and Resources, 2021). However, the economic benefits of mining come at an environmental cost, and the collective footprint of mining in Australia is expected to exceed $4000 \ \mathrm{km}^2$ by 2050 (EPA, 2014), and globally, it is currently $57000 \ \mathrm{km}^2$, and increasing at an unparalleled rate in the last decade. There is little historical evidence of capacity to effectively restore land at this scale (EPA, 2013), and regulatory bodies have urged the mining industry to engage in restoration science (EPA, 2014).

To rehabilitate and restore biodiverse, resilient ecosystems post-mining, we must first measure and diagnose, then reinstate and monitor the health of the soils. However, a lack of rapid, quantitative methods for assessing and monitoring soil health may at least partially underpin the continuing failure to deliver effective and cost-efficient restoration outcomes following mining. In this context, our results are encouraging because they present an opportunity for establishing a science-based diagnostic capacity to rapidly and cost-effectively estimate soil properties that are key for diagnosing soil health. The spectroscopic
models could estimate with at least a moderate or greater accuracy ($\rho_c \geq 0.65$), 24 of the 29 soil physical, chemical, and biological properties tested (Table 5), which represent frequently used (Bünemann et al., 2018; Raghavendra et al., 2020), or recently proposed (Lehmann et al., 2020), indicators of soil health.

In our experiments, linear or polynomial algorithms, PLSR, Cubist, GPRL, and GPRP, resulted in better models (Fig. 3). This might be because of the small size of the dataset. In this case, nonlinear and more complex algorithms are more likely to overfit
and lead to poorer performance. With larger and more diverse data, nonlinear algorithms would produce better predictions (Viscarra Rossel and Behrens, 2010; Tsakiridis et al., 2020).

The accuracy of the estimates of electrical conductivity, $CO_2$ and most of the soil chemical properties from the $A_{350-830}$ spectrometer was markedly poorer than the NIR spectrometers (Fig. 4), indicating that the 350–830 nm range does not hold sufficient chemical information to produce stable models for estimating those soil properties. Combining $A_{350-830}$ with an NIR
spectrometer generated estimates that were similar or better than those made with the portable full-range vis–NIR spectrometer. For instance, the combined $A_{350-830}$ + $C_{1450-2450}$ and $A_{350-830}$ + $D_{1300-2600}$ spectrometers produced more accurate estimates of sand and silt content, $CO_2$, fungal richness, pH, organic C, total N, B, Mn, exchangeable Ca, exchangeable Al and nitrate-N than the full-range spectrometer. Bacterial richness and diversity, P, and Zn and ammonium-N, could not be estimated well with any of the spectrometers or combinations.





The poor repeatability of the $A_{350-830}$ spectrometer's measurements in the 350–500 nm range (Fig. 2) also affected the precision of the spectrometer combinations (Table 4). With a more repeatable visible spectrometer, the spectrometer performance score, $e$, of the combined spectrometers would improve.

How can the miniaturised spectrometers with coarser resolutions and narrower spectral ranges produce nearly as good or better results as the full-range, higher resolution sensor? There might be different reasons. First, soil vis–NIR spectra are non-
specific and highly collinear (Stenberg et al., 2010). Although the miniaturised spectrometers have restricted wavelength ranges, a well-selected spectral range can hold information on overtones and combination bands of important organic and mineral constituents that enable the development of accurate models. For example, spectrometer $B_{1750-2150}$, with a very narrow range, could estimate many soil properties with moderate or greater accuracy ($\rho_c \geq 0.65$, Fig. 4). Table 5 shows the spectrometers with varying spectral ranges that could adequately estimate each soil property. Second, absorptions in soil vis–NIR spectra are
broad (Viscarra Rossel and Behrens, 2010) and coarse spectral resolutions, like those of the miniature spectrometers (Table 2), are unlikely to affect the modelling, compared to the portable vis–NIR spectrometer with a finer spectral resolution.

The portability and affordability (see Table 2) of the miniaturised spectrometers enables the acquisition of soil information at greater temporal and spatial resolutions than conventional laboratory-based methods. Although spectroscopy produces less precise measures of soil properties than conventional laboratory analyses, it enables many more rapid and cost-effective mea-
surements at the appropriate spatial and temporal resolution needed for rehabilitation and ecological restoration. Practitioners can then effectively identify the need for early interventions to establish positive soil health trajectories. In addition, spectroscopy could facilitate the evaluation of soil degradation, more timely identification and remediation of ecologically hostile conditions, and more effective monitoring of the change in soil properties in response to restoration activities.

An additional significance of our results is that the miniaturised spectrometers, even in combinations, are orders of magnitude
cheaper than the full-range portable instrument (Table 2). Our results build on other work that also compares miniaturised spectrometers (Tang et al., 2020; Ng et al., 2020; Sharififar et al., 2019) by testing more sensors, more algorithms for modelling and a more extensive set of soil properties.

Together with other sensors and environmental data, development and further testing of the miniaturised visible and NIR spectrometers could provide the mining industry and restoration practitioners with a rapid and cost-efficient methodology for
diagnosing, assessing, and monitoring soil health. The information gained would ensure that soil management, whether in rehabilitation and restoration, is underpinned by quantitative information for evidence-based decision-making. In addition, the ability to reliably assess a wide range of key soil health indicators enables rapid identification of when intervention is required, which could help deliver significant economic and environmental outcomes.

## 5 Conclusions

Achieving desired outcomes from rehabilitation and ecological restoration activities largely depends upon soil health. Therefore, it is essential to develop efficient, reliable, and cost-effective methods for measuring and monitoring soil properties that can indicate soil health. Our results show that the spectra from miniaturised visible and NIR spectrometers, particularly when

combined, enable the modelling and accurate estimation of many important soil physical, chemical, and biological properties. Importantly, their estimates were as accurate as those from a much more expensive, portable full-range vis–NIR spectrometer. 370 The miniaturised spectrometers allow for the cost-effective acquisition of many more measurements at fine spatial and temporal resolutions, which can improve soil health assessments and, therefore, rehabilitation and ecological restoration outcomes. This information could inform decision-making about the most efficient and effective manner to ameliorate and manage degraded or contaminated soils, delivering significant economic and environmental outcomes.

*Code availability.* The code used for the analyses presented in this work is available from the corresponding author on reasonable request.

*Data availability.* The data used in this work is available from the corresponding author on reasonable request.

*Author contributions.* RVR conceived the study. ZS and RVR performed the data analysis and modelling with input from LW and MZ. HD, KD, PN, designed the soil sampling and HD performed the soil sampling. TMY supplied two of the miniaturised spectrometers and with MM performed the spectroscopic measurements. RVR, ZS led the writing with input from HD, LW, MZ and AC. KD and PN edited versions of the manuscript.

*Competing interests.* The authors have no competing interests to declare.

*Acknowledgements.* Funding to support this project was received from the ARC Centre for Mine Site Restoration, the Research Office at Curtin, Independence Group, Fortescue Metals Group, and a mine in Southwest WA that has requested anonymity. In-kind support was provided by BHP, Alcoa, and Tronox. HD, KD, PN, AC were funded by the Australian Government through the Australian Research Council Industrial Transformation Training Centre for Mine Site Restoration (Project Number ICI150100041). Thank you to Liam Mallon, Grace 385 Abbott, Douglas Laurie, and Jamie Fox for field assistance, Benjamin Moreira Grez and Mieke van der Heyde for molecular work, and Justin Valliere for assistance with determining soil physiochemical properties. We thank trinamiX GmbH (Ludwigshafen, Germany) for supplying us with their Mobile NIR Spectroscopy Solution. The traditional owners of the land on which this research was undertaken are acknowledged and we pay our respects to Elders past, present and emerging.





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
