# Peer review of "Miniaturised visible and near-infrared spectrometers for assessing soil health indicators in mine site rehabilitation"

_SOIL, 2021_

## Community Comment (CC1)

**Discussion on referee 1's comments: Miniaturised visible and near-infrared spectrometers for assessing soil health indicators in mine site rehabilitation by Shen et al.**

We thank the referee for the review and comments. Below we clarify and discuss those comments (in blue text and preceded by **Authors:**). We denote 'manuscript' as 'MS', 'page' as 'P', 'line' as 'L', when referring to locations in the manuscript.

**Comment 1**: The paper number soil-2021-138 of Zefang Shen and co-authors regards the repeatability of the spectrometers and the accuracy of the spectroscopic models built with seven statistical and machine learning algorithms. The authors found that miniaturised spectrometers and combinations predicted 24 of the 29 soil properties with moderate or greater accuracy. The repeatability of the miniaturised NIR spectrometers was similar to that of the full-range, portable spectrometer. The manuscript is potentially a contribution of interest for remote sensing application and it is within its specific scopes but in my opinion the manuscript don't fit in the scope of this journal.

**Authors:** Thank you, the general summary of the paper is correct. However, it is incorrect to suggest that our manuscript pertains to a 'remote sensing application'. Soil visible and near-infrared (vis–NIR) spectroscopy is an analytical technique that is used in the laboratory or in the field. When used in the field, the method is referred to as proximal soil sensing (Viscarra Rossel et al., 2011). Measurements in this same spectral range can be made with remote sensors, from airborne or satellite platforms, but usually with lower spectral resolution and measuring only the very surface of the soil, when it isn't covered by vegetation cover, buildings, etc.(Campbell and Wynne, 2011). The soil samples that we used in our analysis originate from two depths, 0–20 cm and 50–70 cm.

The manuscript fits well in the scope of SOIL, e.g. under "soil protection and remediation" and "soil and methods"

(`https://www.soil-journal.net/about/aims_and_scope.html`), and there are examples of other soil spectroscopic research published in the journal (Baumann et al., 2021; Ng et al., 2020; Yang et al., 2022; England and Viscarra Rossel, 2018). Therefore, we strongly believe our research fits in the scope of SOIL, which includes all topics that fall within the study of soil science as a discipline.

**Comment 2**: In my opinion the work lacks of a clear application of soil, same part of the ms. are very difficult to understand and don't permit to comprehend the originality of the paper.

**Authors:** We are perplexed by the comments that the manuscript lacks a clear soil science application, and that our manuscript lacks originality, but we agree that the description of our experiments are, by the very nature of the technology and analyses, complicated, making parts of the manuscript dense reading.

The 'soil application' pertains to the development of a rapid and cost-effective soil analytical method for the assessment of mining soil and for the purpose of post-mining soil rehabilitation. The conclusions also extend to other soil science applications (i.e. beyond post-mining rehabilitation) where cost-effective measurements of soil properties are required. We feel that this is clearly articulated in the manuscript, particularly in the introduction and discussion.

The originality of our research is summarised in the submitted manuscript (MS P4 L91-93). Of course, if that isn't clear, we can emphasise that in a revision: Our study is the first to evaluate portable, miniaturised vis–NIR spectrometers for assessments of soil in post-mining soil rehabilitation; We compared a diverse set of soil physical, chemical, and biological properties (29 in total) to cover a wide range of soils in different mining contexts; We evaluated miniaturised spectrometers alone and in combinations, which is important as different spectral ranges contain different and potentially unique information; We compared seven different algorithms to demonstrate the robustness of the spectroscopic method; We validated the models using two methods to prevent overly promising results; We considered both the

accuracy and repeatability of the miniaturised spectrometers and their combinations.

We understand that parts of the manuscript are difficult to follow, because our experimental design was complicated by the different comparisons that we made. However, we feel a robust assessment of the method necessitates these comparisons. Thus, in a revision, we propose to improve the readability of the manuscript by including a diagram to summarise the experimental design and experiments conducted. For instance, we could include a diagram like that shown in Figure 1.

[Figure]

Figure 1: Study design

**Comment 3**: As is written do not explain the main results and should be connected with the aim of the research. Instead, the aim (L85-90, pg. 4)) focus only in several approach of multivariate procedure, seems just a mere exercise of multivariate

statistics applied to remote sensing equipment's. .

**Authors:** We assume that the reviewer missed the connection between our aims and results because parts of the manuscript are complex. As we state in our previous discussion point, we propose to improve this in a revision. To clarify, our aim was to assess the repeatability and the accuracy of spectrometers, and their ability to estimate a wide range of soil properties that are considered key indicators of soil health in mine site soil rehabilitation and restoration (MS P3-4, L87-88). To 'connect' our aims and results we used sub-headings in the Results section that directly relate to the aims: Section 3.2 (Assessment of the spectroscopic modelling with the spectrometers, P13) assesses the accuracy of the spectrometers; Section 3.3 (Effect of repeatability on the spectroscopic modelling, P14) assesses the repeatability of the spectrometers via modelling with two spectral replicates respectively; Section 3.4 (overall assessment of the spectrometers, P15) reports the performance of each spectrometer/combination considering both accuracy and repeatability for each of the 29 soil properties. We note also, that we used a similar structure of sub-headings in the Methods for clarity and ease of reference.

We have already clarified the comments around the aims and that this research isn't about 'remote sensing', but here, we must stress that this research isn't only a statistical exercise. Multivariate modelling is fundamental for assessing the soil spectroscopic method. Without the multivariate modelling of the soil properties we could not compare the performance of the spectrometers. We used seven algorithms that account for linear responses (PLSR) to more complex, non-linear responses (e.g. SVM), that also operate and are categorised differently: statistical (PLSR), tree-based (CUBIST, RF, XGBoost), Gaussian process-based (GPRL and GPRR), and support vector methods (SVM). All these algorithms have been employed in literature for soil spectroscopy modelling (Rossel and Behrens, 2010; Liu et al., 2016; Yang et al., 2019; Martínez-España et al., 2019). For a thorough and reliable assessment and comparison, we needed to test these different algorithms. This type of modelling,

and particularly machine learning, is largely dependent on the data set used and there isn't a single 'best' method for all applications. Using only one algorithm could lead to inaccurate conclusions. In a revision, we could of course emphasise the rationale of using the different algorithms, e.g. in section 2.4 Spectroscopic modelling (MS P7).

**Comment 4**: The novelty is not explained as well as the gap in our knowledge that the manuscript with its objectives should fill. For example if the novelty are studies on 1) hyperspectral quantitative analysis (L155 to provide comparison data for the 29 soil chemical, physical and biological soil properties to be assessed using spectroscopic methods) 2) compare Spectral range, resolution, price, weight, and dimensions of miniaturised and portable spectrometers used in this study. (?) , 3) to assess the spectroscopic modelling with different statistical and machine learning algorithms, as well as 4) the accuracy of the spectrometers estimates and their repeatability, 5) assessment of the spectroscopic modelling algorithms (L160) with data from plots or (finally!) a model analysis of physicochemical indicators of polluted soil (?). The gap that results from a state-of-the-art topic should be clarify form the beginning .

**Authors:** Once again, we can only assume that the presentation has made the research difficult to follow and masked the clarity of the knowledge gap our research addresses. In the above discussion points, we have proposed that in a revision, we would improve the clarity of our presentation and description of the experimental design and we have also addressed comments around the aims and originality/novelty of our work. Here, we stress that to robustly assess the use of the miniaturised vis–NIR spectrometers for estimating soil health indicators in mine site rehabilitation, we needed to perform all of the tests and comparisons that we report (most of these are listed by the reviewer in his/her comment points 1–5).

Of course, we can emphasise the knowledge gap that our work addresses. However, we note that in the submitted manuscript (MS, L20-46) we describe context and the knowledge gap by explaining that mine site rehabilitation is guided by the measurement of a range of soil physical, chemical, and biological health indicators

because soil health assessment is key for returning functioning ecosystems, as the soil supports both above- and below-ground biodiversity. Conventional measurement of these indicators is expensive (MS, L52-54). Soil vis–NIR spectroscopy offers a cost-efficient solution for such assessments, and there is growing interest in using miniaturised spectrometers (Kademi et al., 2019; Alcalà et al., 2013; Catelli et al., 2020; Sharififar et al., 2019; Ng et al., 2020; Tang et al., 2020).

**Comment 5**: In my opinion the authors have to clarify in the state of art: 1) why they us this soil health indicators in mine site rehabilitation 2) if the procedure adopted is able to predict the level of contamination or soil health , 3) what are the limits of their predictive model, 4) if the model can be used for other place, because the authors do not compare their results with similar study.

**Authors:** We believe that the submitted manuscript addresses point 1), and we refer the reviewer to the Introduction and cited references that explain the need of a comprehensive set of soil physical, chemical and biological properties (or indicators) for assessments of soil health in mining (Rayment and Lyons, 2010; Munoz-Rojas et al., 2016; Hart et al., 2020).

Regarding point 2), the research presented in this paper focuses on accurately predicting various soil properties that are indicators of soil health. We do not make an assessment of soil health because this is out of scope in this manuscript. With the measured indicators one can derive assessments of soil health, for example, using methods that are proposed in literature (Rinot et al., 2019). We plan to tackle that problem in a following study. Soil health is a complex concept and how it can be accurately estimated is a topic of significant current debate in the literature; its estimation here would only further complicate this research and deviate from the research question, i.e. can we accurately estimating soil health indicators with miniaturised vis–NIR spectrometers to support post-mining soil rehabilitation?

In terms of point 3), we agree that the current Discussion can be enriched by reference to the limitations of the spectroscopic method. Of course, there are

limitations and in a revision, we can elaborate on those.

Regarding point 4), we do not compare our results to similar studies because we have not found any other published research on the use of vis–NIR spectroscopy for mining soil assessments for rehabilitation, or actually on the use of vis–NIR for assessing a diverse range of soil physical, chemical and biological properties. Our results suggest that the miniaturised spectrometers with machine learning can be used to estimate 24 out of 29 soil physical, chemical, and biological properties from different soil types and mining contexts (P4, Table 1). Practitioners can use our results and conclusions to also develop their own models following the sensors and methods we proposed.

**Comment 6**: The second problem of this work is in the preprocessing method that is very confused. The Materials and Methods section (L160-185) does not provide sufficient detail to follow the progress of the manuscript. Regarding methods, there is a use of PLSR, RF, SVM, GBXBoost, CUBIST, SVM, GPRL, GPRP an incredible set of algorithms without explaining the criteria or limits or even if they are designed for these tools. So the reader is assuming or just have to imagine if the spectra were precorcessed. So the row data were filtered with Savitzky-Goaly (SGR) may be with multiplicative scatter correction MSC, or standard normal variate SNV, if they are treated with linear baseline correction LBC, peak normalization N, mean center MC. All this pre-processing are without explanation, whereas all the rest of the methods are only to be found in the results and conclusion discussion section. It is not explained why these methods were used and not others, how they are related to each other, nor is there sufficient detail to understand what was done and how these methods achieved the objectives of the manuscript.

**Authors:** This is a confusing comment. The comment that the 'preprocessing method is very confused' cannot be further from the truth. We described the preprocessing that we used in section '2.3 Soil spectroscopy' (P7, L149-150) and what we did is very simple: 'The reflectance (R) spectra of the soil samples recorded with

each instrument were transformed to apparent absorbance using log10(1/R), and interpolated to 10 nm intervals to attain a consistent wavelength interval.' In our opinion there is nothing confusing about that. We do not use any other preprocessing method. As the reviewer must know, preprocessing methods are data specific and used to correct for errors in the measurements (e.g. either from random noise, or additive, or multiplicative errors). In this case we found no reason to use them. We are perplexed by the reviewer's comment '...All this pre-processing are without explanation, whereas all the rest of the methods are only to be found in the results and conclusion discussion section...' etc. – these and the remaining text in Comment 6 are all obviously incorrect.

Regarding the comment that we do not adequately describe the different algorithms that we used–in a previous discussion point, above, we have already commented on the rationale for using the seven algorithms. Of course, we can provide detail on the algorithms in a revision, however, other authors have described these methods in some detail and have shown how they are suitably used with soil vis–NIR spectra. The manuscript provides references that the interested reader can find to learn more about them. We do not think it necessary to describe the algorithms in this manuscript because others have done so in published papers elsewhere and doing so would unnecessarily lengthen and complicate the manuscript further.

**Comment 7**: I am sorry for the Authors but no revision can at this point improve this work. Many other comments would be possible both for the sections 'Materials and methods' and 'Results and discussion', but it is useless because the comments made are more than sufficient to recommend the rejection of the manuscript.

**Authors:** We hope that our discussion points above clarify and show that the reviewer misunderstood our research and manuscript. We acknowledge that the reporting and presentation of our experiments could be improved, and we have proposed how we might do that in a revision. Finally, we are very concerned about the recommendation to reject our manuscript because as we have shown, the

recommendation is based on significant misunderstandings, misinterpretations and in some instances on comments that appear not to relate to this manuscript.

**References**

Manel Alcalà, Marcelo Blanco, Daniel Moyano, Neville W Broad, Nada O'Brien, Don Friedrich, Frank Pfeifer, and Heinz W Siesler. Qualitative and quantitative pharmaceutical analysis with a novel hand-held miniature near infrared spectrometer. Journal of Near Infrared Spectroscopy, 21(6):445–457, 2013.

Philipp Baumann, Anatol Helfenstein, Andreas Gubler, Armin Keller, Reto Giulio Meuli, Daniel Wächter, Juhwan Lee, Raphael Viscarra Rossel, and Johan Six. Developing the swiss mid-infrared soil spectral library for local estimation and monitoring. Soil, 7(2):525–546, 2021.

James B Campbell and Randolph H Wynne. Introduction to remote sensing. Guilford Press, 2011.

Emilio Catelli, Giorgia Sciutto, Silvia Prati, Marco Valente Chavez Lozano, Lucrezia Gatti, Federico Lugli, Sara Silvestrini, Stefano Benazzi, Emiliano Genorini, and Rocco Mazzeo. A new miniaturised short-wave infrared (swir) spectrometer for on-site cultural heritage investigations. Talanta, 218:121112, 2020.

Jacqueline R England and Raphael A Viscarra Rossel. Proximal sensing for soil carbon accounting. Soil, 4(2):101–122, 2018.

Miranda M. Hart, Adam T. Cross, Haylee M. D'Agui, Kingsley W. Dixon, Mieke Van der Heyde, Bede Mickan, Christina Horst, Benjamin Moreira Grez, Justin M. Valliere, Raphael Viscarra Rossel, Andrew Whiteley, Wei San Wong, Hongtao Zhong, and Paul Nevill. Examining assumptions of soil microbial ecology in the monitoring of ecological restoration. Ecological Solutions and Evidence, 1(2): e12031, 2020. doi: https://doi.org/10.1002/2688-8319.12031. URL https://besjournals.onlinelibrary.wiley.com/doi/abs/10.1002/2688-8319.12031.

Hafizu Ibrahim Kademi, Beyza H Ulusoy, and Canan Hecer. Applications of miniaturized and portable near infrared spectroscopy (nirs) for inspection and control of meat and meat products. Food Reviews International, 35(3):201–220, 2019.

Lanfa Liu, Min Ji, Yunyun Dong, Rongchung Zhang, and Manfred Buchroithner. Quantitative retrieval of organic soil properties from visible near-infrared shortwave infrared (vis-nir-swir) spectroscopy using fractal-based feature extraction. Remote Sensing, 8(12):1035, 2016.

Raquel Martínez-España, Andrés Bueno-Crespo, Jesús Soto, Leslie J Janik, and José M Soriano-Disla. Developing an intelligent system for the prediction of soil properties with a portable mid-infrared instrument. Biosystems Engineering, 177: 101–108, 2019.

Miriam Munoz-Rojas, Todd E. Erickson, Kingsley W. Dixon, and David J. Merritt. Soil quality indicators to assess functionality of restored soils in degraded semiarid ecosystems. Restoration Ecology, 24(August):S43–S52, 2016. doi: 10.1111/rec.12368.

Wartini Ng, Linca Anggria, Adha Fatmah Siregar, Wiwik Hartatik, Yiyi Sulaeman, Edward Jones, Budiman Minasny, et al. Developing a soil spectral library using a low-cost nir spectrometer for precision fertilization in indonesia. Geoderma Regional, 22:e00319, 2020.

GE Rayment and DJ Lyons. Soil Chemical Methods – Australasia. CSIRO Publishing, Canberra, 2010.

Oshri Rinot, Guy J Levy, Yosef Steinberger, Tal Svoray, and Gil Eshel. Soil health assessment: A critical review of current methodologies and a proposed new approach. Science of the Total Environment, 648:1484–1491, 2019.

RA Viscarra Rossel and Thorsten Behrens. Using data mining to model and interpret soil diffuse reflectance spectra. Geoderma, 158(1-2):46–54, 2010.

Amin Sharififar, Kanika Singh, Edward Jones, Frisa Irawan Ginting, and Budiman Minasny. Evaluating a low-cost portable nir spectrometer for the prediction of soil organic and total carbon using different calibration models. Soil Use and Management, 35(4):607–616, 2019.

Yijia Tang, Edward Jones, and Budiman Minasny. Evaluating low-cost portable near infrared sensors for rapid analysis of soils from south eastern australia. Geoderma Regional, 20:e00240, 2020.

RA Viscarra Rossel, VI Adamchuk, KA Sudduth, NJ McKenzie, and Craig Lobsey. Proximal soil sensing: An effective approach for soil measurements in space and time. Advances in agronomy, 113:243–291, 2011.

Y. Yang, Z. Shen, A. Bissett, and R. A. Viscarra Rossel. Estimating soil fungal abundance and diversity at a macroecological scale with deep learning spectrotransfer functions. SOIL, 8(1):223–235, 2022. doi: 10.5194/soil-8-223-2022. URL https://soil.copernicus.org/articles/8/223/2022/.

Yuanyuan Yang, Raphael A Viscarra Rossel, Shuo Li, Andrew Bissett, Juhwan Lee, Zhou Shi, Thorsten Behrens, and Leon Court. Soil bacterial abundance and diversity better explained and predicted with spectro-transfer functions. Soil Biology and Biochemistry, 129:29–38, 2019.

---

## Community Comment (CC2)

**Discussion on referee 2's comments: Miniaturised visible and near-infrared spectrometers for assessing soil health indicators in mine site rehabilitation by Shen et al.**

We thank the referee for the comments. Below we provide a discussion (in blue text and preceded by **Authors:**). We denote 'manuscript' as 'MS', 'page' as 'P', 'line' as 'L', when referring locations in the manuscript.

**Comment 1**: The paper discusses about several statistical and machine learning algorithms for evaluation of the spectrometers and the model prediction accuracy. Many soil physical, chemical and biological properties are targeted. It would give a reference for further NIR application. The paper should be improved before publication. Some suggestions are listed below

**Authors:** We thank the referee for reviewing our manuscript.

**Comment 2**: Please give details of the experimental design for spectroscopy measurement. How can you observe the data in Figure 2.

**Authors:** The experimental design for the spectroscopic measurements are given in the Methods subsection 2.3 Soil spectroscopy and the spectroscopic measurements (P6, L141-148). However, we agree that the description of our experiments and analyses are complicated, making parts of the manuscript difficult to understand. Thus, in a revision, we propose to include a diagram to summarise the experimental design and experiments conducted. For instance, we could include a diagram like that shown in Figure 1.

[Figure]

Figure 1: Study design

One such figure should also help to clarify how we produced our results, including Fig. 2 in the submitted manuscript. For MS Fig. 2, we calculated the mean of the two replicates ($Rep_A$ and $Rep_B$) for each spectrometer, which gives the plots in the first column of Fig. 2 (MS, P11). We also derived the difference between $Rep_A$ and $Rep_B$ using equation (1) (MS, P9) for each sample, which gives the second column in Fig.2. These plots show the repeatability of the spectroscopic measurements for each spectrometer. A smaller difference suggests better repeatability. The third column in Fig. 2 (MS, P11) shows results for the combined spectrometers.

**Comment 3**: I do not think that 56 samples for modeling is enough. Please prove and validate it, or I am not convinced of the results.

**Authors:** In total, we collected, analysed, and measured with the spectrometers 280 soil samples from subplots (described in MS Methods, P5–7). We performed the assessments and modelling in two ways. First, we aggregated the 280 subplots into 56 plots and we performed the modelling and validation on the aggregated data and using 10-fold cross validation (described in sub-section 2.4.1 Assessment of the spectroscopic modelling algorithms with data from plot). We aggregated the data for two reasons, (i) soil samples from a single plot were assumed to be somewhat similar and (ii) computational efficiency, since we assessed in terms of accuracy and repeatability seven spectrometers and combinations using seven algorithms and 29 soil properties, which produced approximately 4263 model evaluations with 10-fold cross validation).

Second, to ensure that our validation with the 56 data from plots was reasonable, we also performed the modelling and validation with the 280 data from the subplots but this time using 10-fold-plot-out cross validation (described in sub-section 2.4.4 Assessment of the spectroscopic modelling with data from subplots). The results from both of these approaches were similar and demonstrate the robustness of the results.

**Comment 4**: The definition of Lin's concordance correlation is not given.

**Authors:** We did provide a reference for the coefficient but in a revision we can of course also include a definition in L185 (MS P8) as follows:

'$\rho_c$ measures the deviation from a 45-degree line of perfect agreement between the observed and predicted values. It ranges from -1 to 1, with 1 denoting perfect agreement.'

**Comment 5**: Analysis of prediction errors (like RMSE) are needed in results and discussions part.

**Authors:** Please note that we did provide analysis of the errors using the root mean squared error (RMSE), mean error (ME), and standard deviation of the error (SDE) (Table 5 in the MS). The RMSE quantifies the inaccuracy, ME the bias, and SDE for

the imprecision such that $RMSE^2 = ME^2 + SDE^2$. And we do discuss these in the Results and Discussion sections. However, in a revision we can further emphasise the implications of these results.

---

## Community Comment (CC3)

**Discussion to referee 1's new comments: Miniaturised visible and near-infrared spectrometers for assessing soil health indicators in mine site rehabilitation by Shen et al.**

Thank you for taking the time to read our comments. We appreciate the opportunity that SOIL provides for an open discussion. We will not respond to comments we have already addressed to keep our discussion brief. Here, we give only further clarifications and supplementary responses.

We do not see a mismatch between our research and the journal's scope, which includes '...all topics that fall within the study of soil science'. Our manuscript pertains to the development of 'soil and methods' for improving 'soil protection and remediation'. Soil vis–NIR spectroscopy is a soil analytical method for estimating soil properties. The assessment of soil health indicators in post-mining soil assessment and remediation are also aspects of soil science relevant to this journal and its readership.

Our manuscript's aims are specific (MS P3-4, L87-88). Therefore, it is out of scope to cover topics related to spatial variability (lateral or vertical), spatial dependence, toxic bioavailability, hierarchical soil classification, soil formation processes, contamination by heavy metals, etc. The reviewer misunderstood. The samples we used were reference or stockpiled native soils; thus, they were not contaminated.

Our sampling method did not 'ignore vertical spatial variability', but in this specific study, we did not think it necessary to characterise the vertical spatial variability of the stockpiles. We sampled reference and stockpiled soils from the 0–20 cm layer. At the youngest stockpile in each mine, we collected additional samples from the 50—70 cm layer, which corresponds roughly to the top layer of the original soil before stockpiling. See section 2.1 Sampling design, which describes the procedure. It was more important to sample soil from across different mines with different soil types and soil conditions to test the applicability of the miniaturised spectrometers and the spectroscopic method using different soils with widely ranging

soil properties.

Of course, the cited references will help to understand the algorithms we used more deeply. We do not describe them in detail because these are published elsewhere, and it makes no sense to paraphrase text from current publications. If readers want the detail, then as is customary, and not unreasonably, we expect them to find the references and read those papers. We note, however, that in the submitted manuscript, we describe our implementation of those algorithms (e.g. the optimisation of the hyperparameters, etc.), which is essential for readers to understand better what we did. Please see section 2.4.1 Assessment of the spectroscopic modelling algorithms...

We have proposed a new figure to illustrate better and clarify the experimental design and methodology. We appreciate the reviewer's feedback that this figure is confusing. In a revision, we could consider this and draw a figure that is easier to understand with a 'higher level' description of what we did.

The reviewer writes that 'this study merely repeats the well-known findings on predictive models', but we have already explained that our study isn't simply a statistical exercise. Furthermore, we reiterate that we have not found publications with a comprehensive comparison like we have done or on the topic of our manuscript.

We strongly disagree that there are 'two fundamental problems' with our spectroscopic modelling ('Chemometric technique performances'). The reviewer suggests that one of these problems is that we 'compare soil properties only on the basis of depth', but nowhere in our manuscript do we write this because it is incorrect. The second problem '...that the paper does not discuss convincingly the limitations of the approach and potential biases due to the assumptions made...' isn't a fundamental problem of the spectroscopic modelling. We have already acknowledged (see our previous discussion Comment 5) that in a revision, we could enrich the manuscript by elaborating on the limitations of the spectroscopic method.

We once more thank the referee for reading and reviewing the manuscript, but

we remain perplexed by many of the comments made. The reviewer states that the manuscript is well-written, so we can only assume that his/her serious misunderstandings are due to our complex descriptions of the technology and analyses, which make parts of the manuscript dense reading. We are committed to improving our manuscript in a revision as we have previously proposed.

---

## Author Response (AR2)

**Changes made for production**

June 30, 2022

We have made some minor changes to the manuscript, listed as below:

- We added photos of the spectrometers in Fig. 5 (P14). Please note that the photos were entirely created by the authors. The results indicated by the figure remain the same.

- The order of the co-authors has also been changed compared with the initial submission.

- We corrected typos, improved grammar, and modified Conclusions for better readability.